# Improved RAkEL’s Fault Diagnosis Method for High-Speed Train Traction Transformer

**DOI:** 10.3390/s23198067

**Published:** 2023-09-25

**Authors:** Man Li, Xinyi Zhou, Siyao Qin, Ziyan Bin, Yanhui Wang

**Affiliations:** 1State Key Laboratory of Advanced Rail Autonomous Operation, Beijing Jiaotong University, Beijing 100044, China; 21120988@bjtu.edu.cn (X.Z.); wangyanhui@bjtu.edu.cn (Y.W.); 2School of Traffic and Transportation, Beijing Jiaotong University, Beijing 100044, China; 20251047@bjtu.edu.cn (S.Q.); 20251031@bjtu.edu.cn (Z.B.); 3Beijing Research Center of Urban Traffic Information Sensing and Service Technology, Beijing Jiaotong University, Beijing 100044, China

**Keywords:** multi-label classification, RAkEL, traction transformer, fault diagnosis

## Abstract

The traction system is very important to ensure the safe operation of high-speed trains, and the failure of the traction transformer is the most likely fault in the traction system. Fault diagnosis in actual work relies largely on manual experience. This paper proposes an improved RAkEL (Random *k*-Labelsets) algorithm for the fault diagnosis of high-speed train traction transformers. Firstly, this article starts from the large amount of “sleeping” fault maintenance data accumulated by the railway department, takes a single maintenance record as an instance, uses specific monitoring values to construct an instance vector, and uses the fault phenomena corresponding to the monitoring indicators as labels. Then, this paper improves the step of selecting *k*-labelsets in RAkEL, and extracts associated faults using the Relief algorithm. Finally, this paper excavates and uses the association rules between data and faults to identify traction transformer faults. The results showed that the improved RAkEL diagnostic method had a significant improvement in the evaluation indicators. Compared with other multi-label classification algorithms, including BR (Binary Relevance) and CLR (Calibrated Label Ranking), this method performs well on multiple evaluation indicators. It can further help engineers perform timely maintenance work in the future.

## 1. Introduction

In recent years, high-speed railway trains have developed rapidly in terms of speed and capacity. The system structure is complex, the equipment is closely connected with each other, and there are many types of faults. An efficient and accurate fault diagnosis mechanism is of great significance to reducing or eliminating the occurrence of accidents and ensuring the safety of high-speed train operations. The traction system is a key subsystem of high-speed trains, and the traction transformer is one of the main equipment. It is composed of iron core, coil winding, oil tank, oil protection device, and other equipment. Its operation is often affected by various factors such as electricity, heat, and magnetism. With the increase in service time, changeable train operating environment, or due to defects in the manufacturing process, irregular maintenance work, etc. [1], various failures of traction transformers will inevitably occur, burying potential safety hazards.

The large amount of maintenance data accumulated by the railway department can provide data support for fault diagnosis. The data-driven fault diagnosis method has the advantages of a wide application range, less need for modeling and prior knowledge, and easy implementation [1]. In the field of data-driven fault diagnosis of complex electromechanical equipment, machine learning methods are widely used due to their high reliability and easy implementation [2,3,4,5,6]. For example, support vector machine (SVM), a new machine learning method based on statistical learning theory, is a powerful tool for solving small sampling, nonlinear, and high-dimensional problems, which has inspired several works in recent years [7,8]. Many scholars optimize penalty factors and kernel function parameters through intelligent optimization algorithms, such as applying particle swarm optimization (PSO) [9], improved seagull optimization algorithm [10], etc., to improve diagnostic accuracy. Bayesian networks are also widely used in this field [11,12].

The key to fault diagnosis is feature extraction. The principal component analysis (PCA) method is another important research topic in the field of system fault analysis [13,14]. Among them, kernel principal component analysis (KPCA) is a commonly used method, such as the fuzzy clustering-based operating state diagnosis algorithm proposed in reference [15], or by combining it with SVM [10] to further obtain fault information. Deep PCA combines PCA theory with deep learning to effectively achieve early fault diagnosis [16,17].

In the diagnosis of faults in complex electromechanical equipment, existing research predominantly relies on Dissolved Gas Analysis (DGA), processing its characteristic values, and identifying corresponding single-fault types. However, these studies often overlook the fact that in practical operation, equipment instances are often associated with one or more fault categories, naturally giving rise to multi-label classification problems. Therefore, in recent years, multi-label classification methods have become increasingly popular in the field of fault diagnosis (three articles), which can well explore fault diagnosis situations of high-dimensional features and compound fault types. There are currently a large number of mature methods in the field to solve multi-label learning problems [18]. The first is the transformation method problem, such as BR (Binary Correlation), CC (Classifier Chains), CLR (Calibrated Label Ranking) [19,20,21]. The other is to improve the existing single classifier, such as transforming the nearest neighbor classifier (KNN) into multi-label nearest neighbor classifier (MLKNN), transforming support vector machine (SVM) into rank support vector machine (RANKSVM) [22,23]. The classic multi-label classification algorithm RAkEL has been widely used in image classification [24], biomedicine [25,26], information security, and other fields [27]. By dividing the label space, RAkEL tries to overcome the problems of computational cost and label powerset explosion. This paper is based on the ordinary RAkEL method, treats labels as features, uses the Relief algorithm to improve the steps of randomly selecting labelsets, and diagnoses high-speed train traction transformer faults based on existing fault and maintenance data. The results show that, compared with the ordinary RAkEL and other ordinary multi-label classification algorithms, the improved RAkEL has better performance.

The main contributions of this paper are as follows:1.Using the improved RAkEL method proposed in this paper to mine the correlation between fault phenomena in the actual high-speed train traction transformer fault dataset, the accuracy of the fault phenomenon identification and final maintenance diagnosis is high. At the same time, this paper sets relevant parameters based on the high-speed train traction transformer dataset and achieves good diagnostic results, indicating that this method is suitable for the actual fault diagnosis process.2.Based on the ordinary RAkEL algorithm, this paper considers the correlation between fault manifestations in the process of selecting *k*-labelsets, adds the Relief algorithm to reduce the randomness of labelset selection, and improves the calculation efficiency and diagnosis accuracy. In the actual high-speed train traction transformer fault dataset, compared with the ordinary RAkEL, the improved RAkEL has better performance in various evaluation indicators. Based on the optimal parameters obtained through experiments, AP
increased by 7.4%, and Coverage, Hamming Loss, One Error, and Ranking Loss decreased by 51.2%, 9.8%, 13.3%, and 51.6%, respectively.3.After adding the Relief algorithm to mine label correlation, the improved RAkEL performs the best overall in comparison with other algorithms. In the actual high-speed train traction transformer fault dataset, based on the set parameters, the improved RAkEL has the best comprehensive performance compared with BR, CLR, and LP, and is only slightly lower than BR in AP. This shows that compared with BR, the number of related instances that have not been diagnosed is larger in the improved RAkEL method.

The organizational structure of the rest of the paper is as follows: Section 2 introduces the working content and diagnostic difficulties of the traction transformer and the RAkEL method. Section 3 introduces the model construction, including the data processing process and evaluation indicator system. Section 4 uses the known data to verify the calculation examples, and discusses the related parameters. Section 5 summarizes the full text, and puts forward the deficiencies and the direction of future efforts.

## 2. Background and Related Work

### 2.1. Work Content and Diagnostic Difficulties of Traction Transformers

Taking the CRH5 model as an example, the multiple unit is composed of eight car formations and two power units. The first power unit consists of three high-speed trains and one trailer (M-M-T-M); The second power unit consists of two high-speed trains and two trailers (T-T-M-M). Each power unit is equipped with a main transformer (TT) and a pantograph, and the entire train is equipped with two pantographs.

The design of the traction structure is that the third traction converter of the first traction unit can be powered by the second traction unit and can be switched from the first traction unit to another traction unit. This feature enables the balancing of two main transformers when a traction converter fails; at least three traction converters operate when one main transformer fails. When the electrical equipment in each traction power unit malfunctions, the power train of that unit can be completely or partially cut off (cutting off one or two traction converters), but it does not affect the operation of other power units.

The main transformer (TT) is a single pole transformer equipped with six secondary windings, which allows the train to operate on a line powered by a nominal AC voltage of 25 kV–50 Hz, reducing the voltage to a value suitable for driving the parts. It is cooled by forced oil circulation; a dedicated oil-gas heat exchanger (cooler) is used to cool the oil, and an oil evaporator is also integrated into the transformer to ensure the space required for evaporation and oil storage. The main transformer (TT) meets the needs of the traction unit; each traction/auxiliary converter is powered by two secondary windings. In the component called “HV Controller Box” (COMB) adjacent to the transformer, remote isolation switches SAZ1 and SAZ (to drive one or eight vehicles, respectively), SAZ21 and SAZ122 (to drive two or seven vehicles, respectively), and SAZ31 and SAZ32 (to drive four vehicles) are installed to perform the task of disconnecting from the traction converter in the event of a fault affecting the converter.

Figure 1 shows the current phase measurement points for traction transformers, such as breakdown voltage (BV), moisture content, acid value, and dielectric loss factor. Due to the complex types of faults in traction transformers, there is currently little research on the analysis of the relationship between faults in traction transformers. Currently, maintenance work mainly relies on manual testing and judgment. Workers often misjudge or miss faults due to lack of experience or misjudgment, which seriously affects the safe operation of equipment. Therefore, analyzing the relationship between closely related faults and establishing an efficient and accurate fault diagnosis model is crucial for improving the reliability of traction transformer equipment.

### 2.2. RAkEL

The RAkEL method proposed by Tsoumakas et al. involves training multi-classifiers for each *k*-labelset using the LP method after randomly dividing the total labelset of the data into multiple small labelsets containing *k* labels (*k*-labelsets), collecting and combining the decisions of all the LP classifiers to constitute a multi-label classification of unlabeled instances [28]. Compared to LP, RAkEL has the advantage of generating computationally simpler single-label classification tasks with a more balanced distribution of class values. And in the case of overlapping labelsets, RAkEL is able to collect multiple diagnostic results for the same label through different LP models, which are voted to obtain the final output. This provides diagnostic results with the opportunity to correct potentially irrelevant errors and improve performance. At the same time, RAkEL can generalize to labelsets other than the known ones, which LP cannot do. The pseudocode of the algorithm is shown in Algorithm 1.
**Algorithm 1.** Pseudocode of RAkEL.$Y=RAkEL\left(D,M,k,m,x\right)$1.  **for** *r* = 1 to m **do:**2.   Randomly select a *k*-labelset $L^{k}(l_{r})\subseteq L$ with $|L^{k}\left(l_{r}\right)|=k$;3.   Construct the multi-class training set $D_{L^{k}\left(l_{r}\right)}^{+}$ according to Equation (8);4.   $g_{L^{k}\left(l_{r}\right)}^{+}⟵M\left(D_{L^{k}\left(l_{r}\right)}^{+}\right)$;5.  **end for**6.  Return the diagnostic labelset Y according to Equation (15).

## 3. Model Construction

At this stage, the railway department has accumulated a large amount of high-speed train traction transformer operation and maintenance data, including condition monitoring data, fault types and performance, repair and maintenance records, etc. It is of great significance to analyze and excavate these dormant data for the traction transformer’s fault identification and diagnosis of the repair method.

The ordinary RAkEL method randomly decomposes the initial labelset into multiple small labelsets, which reduces the complexity of the generated single-label classification task, but also brings about the problems of redundancy and low computational efficiency. Therefore, this paper considers adding the Relief algorithm for improvement, treating labels as features, and preferentially selecting relevant labels to form a labelset. Finally, we use the improved RAkEL algorithm to mine the association rules between data and faults, faults and faults, point out the types of faults that occur in the newly generated data, and further provide preventive maintenance recommendations.

### 3.1. Constructing the Training Set Containing Instances of Maintenance Records and Corresponding Fault Phenomenon Labels

The historical condition-monitoring data in the existing records contain monitoring values for specific monitoring indicators, including BV, moisture content, dielectric loss factor, and so on. In this paper, we take a single maintenance record as an instance and use its corresponding specific monitoring values to construct the instance space in the multi-label classification algorithm:(1)$$X=\left\{x_{1},x_{2}\ldots x_{i}\ldots x_{N}\right\},$$
where X denotes the instance space and $x_{i}$ denotes the *i*th maintenance record as the instance. $x_{i}$ is a *d*-dimensional feature vector with *d* corresponding to the total number of monitoring indicators.

In this paper, the fault phenomena corresponding to the monitoring indicators are used as labels to construct the label space in the multi-label classification algorithm:(2)$$L=\left\{y_{1},y_{2}\ldots y_{M}\right\},$$
where L denotes the label space, *M* is the total number of possible fault phenomena, and $y_{M}$ denotes the *M*th label contained in the label space. Based on the fault phenomena in each maintenance record, the label 0/1 matrix corresponding to the instance is constructed:(3)$$Y_{i}=(a_{1},a_{2},\ldots {,a}_{j},\ldots {,a}_{M}),$$
$Y_{i}$ denotes the labelset associated with the *i*th instance $x_{i}$, where i=1…N. $a_{j}$ indicates whether the *j*th fault phenomenon occurs or not, if the fault occurs, then $a_{j}$ = 1, otherwise, it is 0, where j=1…M.

The training set containing maintenance record instances and their associated labelsets is denoted as:(4)$$D=\left\{\left(x_{i},Y_{i}\right)\left|1\leq i\leq N\right.\right\},$$
D denotes the training set, where $x_{i}\in X$ denotes the *i*th maintenance record instance, $Y_{i}\subseteq L$ is the number of instances associated with the set of related labels $x_{i}$, and *N* is the total number of instances.

### 3.2. Constructing Multi-Classification Training Sets

RAkEL is an algorithm that transforms a multi-label problem into a multi-classification problem. It reduces the label space into *k*-labelsets and calls LP methods separately for each *k*-labelset to train the multi-classifiers and finally votes to obtain the results to improve the accuracy.

Given the labelset size *k*, the set consisting of the labels in the space of *k* labels is called a *k*-labelset. We construct the set of all possible *k*-labelsets:(5)Lk=Mk,
where $L^{k}$ is the set of all possible *k*-labelsets, $\left|L^{k}\right|$ denotes the size of $L^{k}$, *M* is the number of labels, and Mk denotes that *k* labels are randomly taken from *M* labels.

For the labelsets in $L^{k}$:(6)Lkl=k,1≤l≤Mk,
where $L^{k}\left(l\right)$ denotes the *l*th labelset in $L^{k}$ and $\left|L^{k}\left(l\right)\right|$ denotes the size of the $L^{k}\left(l\right)$.

Given the desired number of classifiers *m*:(7)$$m\leq \left|L^{k}\right|.$$

In the training phase, for an original multi-labeled sub-training set, we reduce the original labeling space L to $L^{k}\left(l\right)$ by converting it to the following multi-class single-labeled training set:(8)$$D_{L^{k}\left(l\right)}^{+}=\left\{\left(x_{i},\sigma_{L^{k}\left(l\right)}\left(Y_{i}\cap L^{k}\left(l\right)\right)\right)|1\leq i\leq N\right\}.$$

$D_{L^{k}\left(l\right)}^{+}$ contains the new class:(9)$$\Gamma \left(D_{L^{k}\left(l\right)}^{+}\right)=\left\{\sigma_{L^{k}\left(l\right)}\left(Y_{i}\cap L^{k}\left(l\right)\right)|1\leq i\leq N\right\},$$
where $D_{L^{k}\left(l\right)}^{+}$ denotes the training set with label space $L^{k}\left(l\right)$; $\sigma_{L^{k}\left(l\right)}:2^{L}\to N$ is the mapping from the power set of L to the natural numbers of the inverse function; and $\Gamma \left(D_{L^{k}\left(l\right)}^{+}\right)$ denotes the new class in $D_{L^{k}\left(l\right)}^{+}$.

### 3.3. Constructing a Collection of Multi-Class Classifiers

A multi-class learning algorithm is utilized to induce a multi-class classifier:(10)$$g_{L^{k}\left(l\right)}^{+}:x\to \Gamma \left(D_{L^{k}\left(l\right)}^{+}\right),$$
i.e.,
(11)$$g_{L^{k}\left(l\right)}^{+}\leftarrow M\left(D_{L^{k}\left(l\right)}^{+}\right),$$

To create an integration with *m* component classifiers, a collection of multi-label classifiers is constructed by calling LP on a set of *m* randomly chosen *k*-labelsets $L^{k}(l_{r})$ (1 *≤ r ≤ m*):(12)$$\overset{⃑}{G}=\left\{g_{L^{k}\left(l_{1}\right)}^{+},g_{L^{k}\left(l_{2}\right)}^{+}\ldots g_{L^{k}\left(l_{m}\right)}^{+}\right\},$$
where $g_{L^{k}\left(l_{m}\right)}^{+}$ denotes the multi-class classifier constructed for the *m*th *k*-labelset.

The ordinary RAkEL randomly selects labels to divide the labelset with strong randomness and ignores the correlation between labels. It is easy to select a large number of redundant or irrelevant label combinations during random selection, which reduces the computational efficiency and diagnostic accuracy. Based on this, this paper applies the *Relief* algorithm, which treats labels as features and constructs closely related labelsets to reduce randomness [29].

Finally, we construct the related label matrix.
(13)$$A_{j}=\{(a_{i},\Phi (Y_{i},y_{j}))|1\leq i\leq N\},$$
(14)$$a_{i}=[\Phi (Y_{i},y_{1}^{c}),\Phi (Y_{i},y_{2}^{c}),\ldots \ldots ,\Phi (Y_{i},y_{j}^{c})],$$
where $A_{j}$ denotes the related label matrix of the *j*th label, $a_{i}$ represents the value of $x_{i}$ with respect to the label in $Y_{j}^{C}$, *N* denotes the number of instances in the training set, $Y_{j}^{C}$ denotes the label matrix consisting of labels other than the *j*th label, $\Phi (Y_{i},y_{j})$ denotes the value of $x_{i}$ with respect to the label $y_{j}$, and $\Phi (Y_{i},y_{j})$ is 1 if $y_{j}\in Y_{i}$ and 0 otherwise. $Y_{i}$ denotes the set of related labels associated with $x_{i}$; $y_{1}^{c}$ denotes the first label in $Y_{j}^{C}$; and $y_{k}^{c}$ denotes the *j*th label in $Y_{j}^{C}$.

We apply the *Relief* algorithm to $A_{j}$ to obtain the weight of each of the remaining labels on $y_{j}$, and next take the *k* − 1 label with the largest weight value among them as the closely related label of the *j*th label to obtain the set of closely related labels for each label $R_{j}$:(15)$$R_{j}=\{y_{b}|rank(y_{b})\leq k-1,y_{b}\in Y_{j}^{C}\},$$
where $R_{j}$ denotes the set of closely related label corresponding to the *j*th label; $rank(y_{b})$ denotes the ranking of weight values corresponding to the *b*th label in $Y_{j}^{C}$.

By forming a *k*-labelset with the *j*th label and its closely related labelset, we obtain a total of *M* closely related *k*-labelsets, and then randomly select *m-M* non-repeated *k*-labelsets from them to jointly construct a multi-classifier ensemble, which reduces the randomness in the selection of *k*-labelsets.

### 3.4. Building Diagnostic Instances Labelsets

For unrecognized record instances x, the following two quantities are calculated for each label:(16)$$\tau (x,y_{j})=\sum_{r=1}^{m}⟦y_{j}\in L^{k}\left(l_{r}\right)⟧\left(1\leq j\leq M\right),$$
(17)$$\mu (x,y_{j})=\sum_{r=1}^{m}⟦y_{j}\in \sigma_{L^{k}\left(l_{r}\right)}^{-1}\left(g_{L^{k}\left(l_{r}\right)}^{+}\left(x\right)\right)⟧,$$
where $\tau (x,y_{j})$ counts the maximum number of votes for diagnostic integration on label $y_{j}$ and $\mu (x,y_{j})$ counts the actual number of votes for diagnostic integration on label $y_{j}$. $⟦X⟧$ denotes the voting method, $⟦X⟧$ is 1 if *X* is true, and 0 otherwise. $\sigma_{L^{k}\left(l\right)}:2^{L}\to N$ is the mapping from the power set of L to the natural numbers of the inverse function; $\sigma_{L^{k}\left(l\right)}^{-1}$ represents the inverse function corresponding to it. $g_{L^{k}\left(l_{r}\right)}^{+}$ denotes the multi-class classifier constructed for the *r*th k-labelset.

Then, the set of diagnostic labels for undiagnosed instances is represented as follows:(18)$$Y=\left\{y_{j}\left|\mu \left(x,y_{j}\right)/\tau \left(x,y_{j}\right)>0.5,1\leq j\leq M\right.\right\},$$

In other words, it is considered that $y_{j}$ is related to x when the actual number of votes is more than half of the maximum number of votes. For a collection created from a set of k labels, the maximum number of votes on each label is mk/M on average.

Table 1 shows the diagnostic voting process of RAkEL for a multi-label training set with the number of labels *M* = 6, for example, run with the labelset size *k* = 3 and the number of desired classifiers *m* = 7.

### 3.5. Evaluation Indicators System

#### 3.5.1. Hamming Loss

*Hamming Loss* is used to evaluate how often labels are misdiagnosed, recording cases where: relevant labels are misdiagnosed as irrelevant, and irrelevant labels are misdiagnosed as relevant.
(19)$$Hamming\;Loss=\frac{1}{p}\sum_{i=1}^{p}\frac{1}{q}|h(x_{i})|,$$
where *p* denotes the number of instances, *q* is the number of labels, and $h(x_{i})$ denotes the number of labels corresponding to the diagnosed error for instance $x_{i}$.

#### 3.5.2. Ranking Loss

*Ranking Loss* evaluates the proportion of reverse-ordered label pairs, recording the proportion of instances where the diagnostic rank of relevant labels is lower than the diagnostic rank of irrelevant labels.
(20)$$rankingloss=\frac{1}{p}\sum_{i=1}^{p}\frac{1}{\left|Y_{i}\right|\left|{\bar{Y}}_{i}\right|}|\{(l’,l”)|f(x_{i},l’)\leq f(x_{i},l”),)l’,l”)\in Y_{i}\times {\bar{Y}}_{i}\}|,$$
where l’ denotes the actual relevant label for the *i*th instance and $l^{\prime \prime }$ denotes the actual irrelevant label for that instance; ${\bar{Y}}_{i}$ is the set of irrelevant labels for the instance; f)x,y) is a real-valued function indicating the confidence that *y* is the relevant label for *x*, obtained by the classifier system.

#### 3.5.3. One Error

One Error is used to evaluate the proportion of instances where the highest-ranked label is not relevant to the instance.
(21)$$oneerror=\frac{1}{p}\sum_{i=1}^{p}\left\{[{argmax}_{l\in y}f(x_{i},l)]\notin Y_{i}\right\},$$
where oneerror(k) is the One Error value of the *k*th label. ${argmax}_{l\in y}f(x_{i},l)$ denotes the highest ranked label associated with instance $x_{i}$.

#### 3.5.4. Coverage

Coverage evaluates how many steps it takes, on average, to move down the list of ranking labels to cover all relevant labels for the instance.
(22)$$coverage=\frac{1}{p}\sum_{i=1}^{p}{max}_{l\in Y_{i}}{rank}_{f}(x_{i},l)-1,$$
where ${rank}_{f}$ is the rank function corresponding to the real-valued function f.

#### 3.5.5. Average Precision

Average Precision (AP) is used to evaluate the average percentage of relevant labels that rank higher than a specific label.
(23)$$avgprec=\frac{1}{p}\sum_{i=1}^{p}\frac{1}{\left|Y_{i}\right|}\sum_{l\in Y_{i}}\frac{\left|\left\{l^{\prime }|{rank}_{f}(x_{i},l^{\prime })\leq {rank}_{f}(x_{i},l),l^{\prime }\in Y_{i}\right\}\right|}{{rank}_{f}(x_{i},l)},$$
where ${rank}_{f}(x_{i},l)$ is the rank function corresponding to the real-valued function indicating the rank of l in the diagnostic results of the unknown instance $x_{i}$.

## 4. Algorithm Validation

The dataset used in this paper is the historical maintenance record dataset of traction transformer for high-speed trains. The maintenance record contains condition monitoring data under different monitoring indicators several times, and the monitoring indicators include BV, moisture content, acid value, etc. Specific monitoring values can reflect certain fault phenomena, and different combinations of fault manifestations can provide recommendations for high-speed train traction transformer maintenance methods.

### 4.1. Parameter Selection

In the RAkEL algorithm, the settings of the labelset size k and the number of classifiers m have a large impact on the diagnostic results. In this paper, 70% is randomly selected from the dataset as the training set and 30% as the test set, and the values of k and m are discussed.

Based on the different values of the k set (*m* = 2 *M*), the change of each evaluation indicator is shown in Figure 2, where coverage has been normalized.

With different values of m set (*k* = 2), the changes in each evaluation indicator are shown in Figure 3, where coverage has been normalized.

With different values of m set (*k* = 3), the changes in each evaluation indicator are shown in Figure 4, where coverage has been normalized:

With different values of m set (*k* = 4), the changes in each evaluation indicator are shown in Figure 5, where coverage has been normalized.

From Figure 3, Figure 4 and Figure 5, it can be seen that the comprehensive performance is good when *k* is set to 3 and m is set to 34, and the results of the indicators near the optimal value show a smooth trend. Considering the test results and suggestions [28], the labelset size *k* is determined as 3 and the number of classifiers *m* is determined as 34.

### 4.2. Comparison of Evaluation Indicators before and after Improvement

In order to reduce the impact of randomness in RAkEL, this paper adds the Relief algorithm that mines label correlation, selects labels with greater correlation to form a labelset, and reduces the redundant and irrelevant impact of randomly selecting a labelset. In this paper, 70% of the dataset is randomly selected as the training set and 30% as the test set; based on the parameters set in the previous section, the indicators before and after RAkEL improvement are shown in Table 2, among which are Hamming Loss, One Error, Coverage, and Ranking Loss, whereby the smaller the value of the indicator, the better the performance, and the larger the value of the AP indicator, the better the performance.

As shown in Figure 6, the improved RAkEL outperforms the ordinary RAkEL in all indicators, with an AP improvement of 7.4%. This indicates that the improved RAkEL algorithm proposed in this paper, which incorporates feature selection and calculates the correlation between labels to prioritize selecting relevant labels for composing *k*-labelsets, has analyzed the association between faults and their concurrent failures effectively. It significantly reduces the impact caused by the random selection of labelsets in ordinary RAkEL, thus validating the effectiveness of the improvement.

### 4.3. Comparison with Other Multi-Label Classification Algorithms

According to the dataset and parameters set in the previous section, Table 3 demonstrates the results of comparing the indicators of this experiment with other multi-label classification algorithms, including BR and CLR, among which are Hamming Loss, One Error, Coverage, and Ranking Loss, whereby the smaller the value of the indicator, the better the performance; the larger the value of the AP indicator, the better the performance.

From Table 3, the improved RAkEL had the best overall performance in all the indicators, except for the slightly lower AP compared to BR.

The comparison of the improved RAkEL indicators with other methods is shown in Figure 7.

As shown in the figure, the improved RAkEL has the best comprehensive performance compared to other classification algorithms in the application of a high-speed train traction system’s fault dataset. This indicates that the improved RAkEL has a better performance in mining the correlation between fault phenomenon labels, and also shows the effectiveness and applicability of mining the correlation of fault phenomena for fault diagnosis and taking effective maintenance measures. The actual fault diagnosis results are shown in Table 4.

In this case, if a breakdown voltage > 50 kV is detected, the probability of diagnosing a C3-level repair is 34.8%; a C4-level repair, 52.2%; and a C6-level repair, 13.0%. The diagnostic accuracy of the final maintenance method reached 93.62%, which is 8.62% higher than the commonly used diagnostic method based on the BPLN model in current engineering, indicating that the method proposed in this paper has practical application value [30].

## 5. Conclusions

It is very important to carry out efficient and accurate fault prediction for the main transformer, as it is one of the components of the high-speed train traction system with the most types of faults and the most frequent monitoring of the phase points. The algorithm proposed in this paper has a small degree of AP and enhancement before and after improvement, and there is still some room for improvement in the evaluation indicators, such as AP, in comparison with other algorithms. The next step we propose is to study the way with which we can achieve a higher-accuracy fault diagnosis while maintaining the existing level.

## Figures and Tables

**Figure 1 sensors-23-08067-f001:**
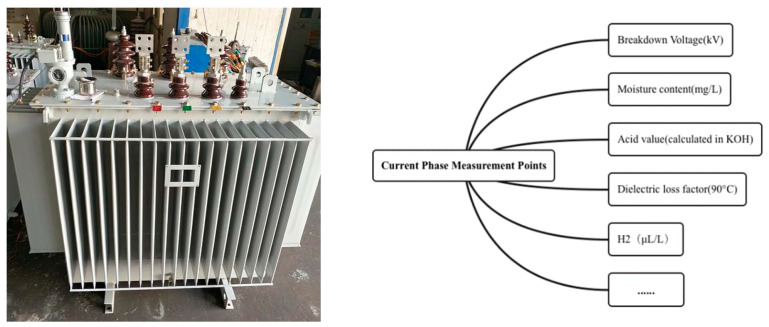
Actual traction transformer and measured phase points.

**Figure 2 sensors-23-08067-f002:**
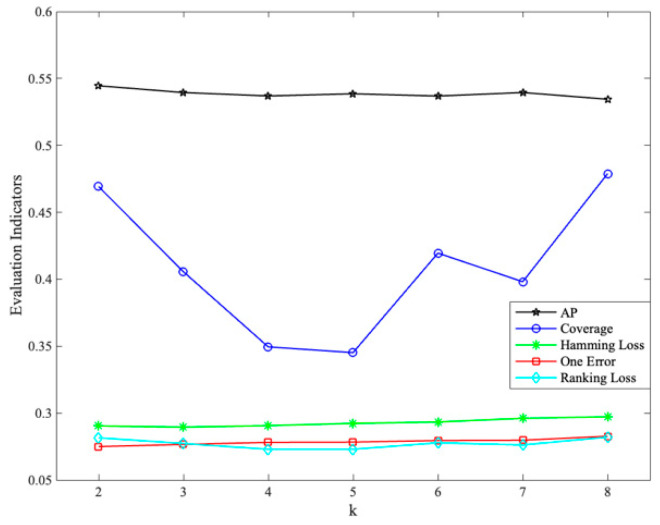
Impact of k setting on evaluation indicators (*m* = 34).

**Figure 3 sensors-23-08067-f003:**
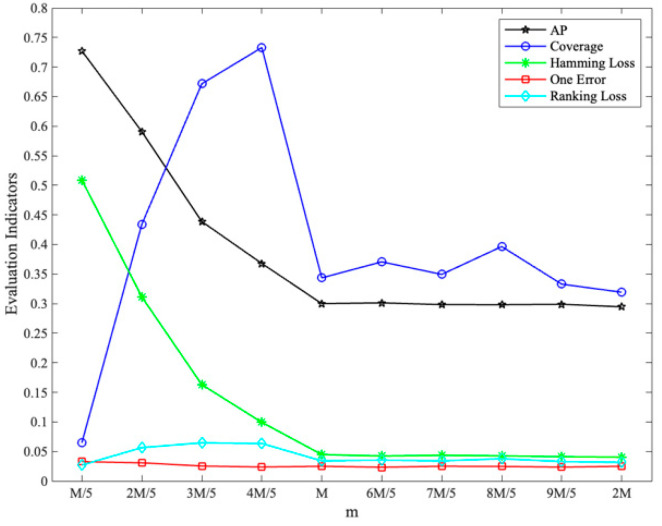
Impact of m setting on evaluation indicators (*k* = 2).

**Figure 4 sensors-23-08067-f004:**
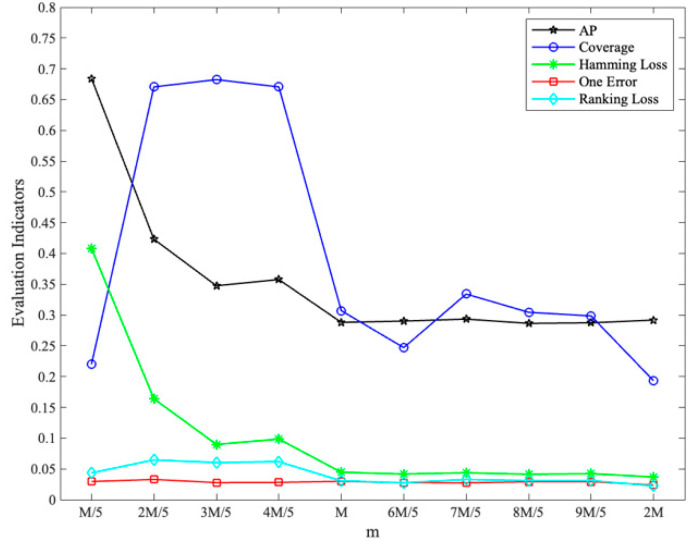
Impact of m setting on evaluation indicators (*k* = 3).

**Figure 5 sensors-23-08067-f005:**
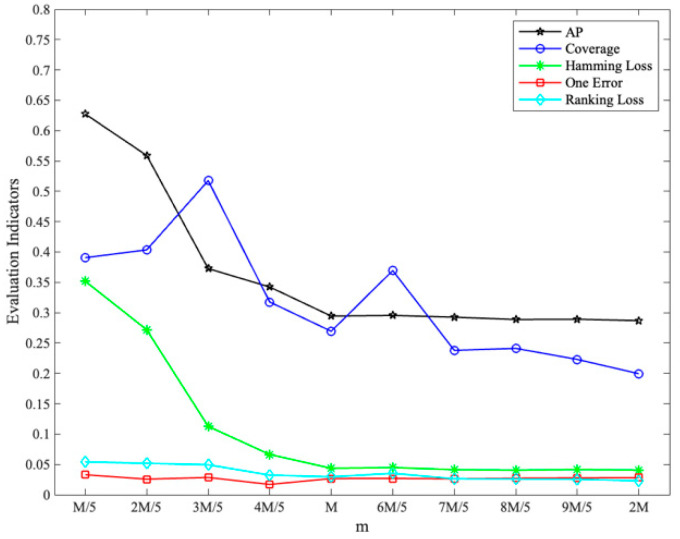
Impact of m setting on evaluation indicators (*k* = 4).

**Figure 6 sensors-23-08067-f006:**
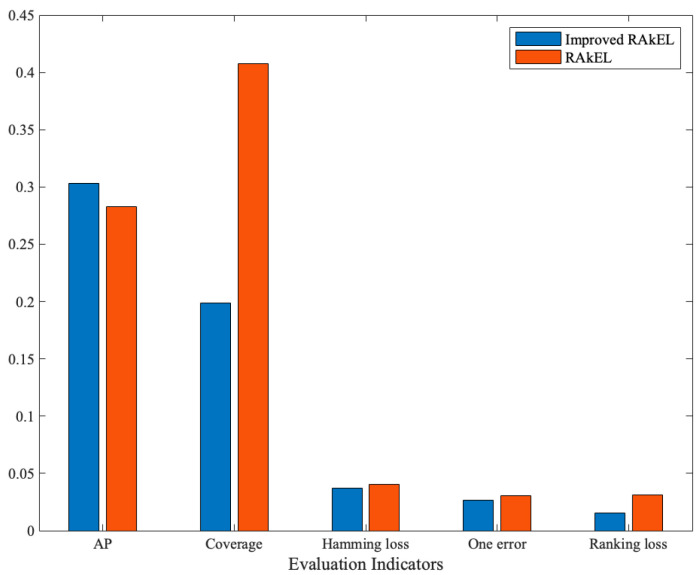
Changes in evaluation indicators before and after RAkEL improvement.

**Figure 7 sensors-23-08067-f007:**
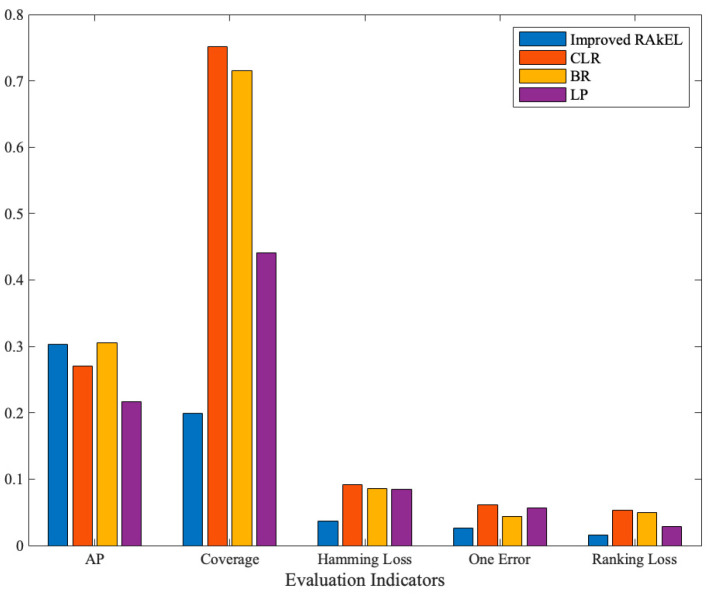
Comparison of improved RAkEL with other algorithms.

**Table 1 sensors-23-08067-t001:** Example of voting process for RAkEL.

Classifier	*k*-Labelset	Diagnostic Labelset					
		$y_{1}$	$y_{2}$	$y_{3}$	$y_{4}$	$y_{5}$	$y_{6}$
$g_{L^{k}\left(l_{1}\right)}^{+}$	$y_{1},y_{2}$ $,y_{3}$	1	0	1	-	-	-
$g_{L^{k}\left(l_{2}\right)}^{+}$	$y_{2},y_{3}$ $,y_{5}$	-	1	0	-	1	-
$g_{L^{k}\left(l_{3}\right)}^{+}$	$y_{3},y_{4}$ $,y_{6}$	-	-	1	0	-	0
$g_{L^{k}\left(l_{4}\right)}^{+}$	$y_{2},y_{4}$ $,y_{6}$	-	0	-	1	-	1
$g_{L^{k}\left(l_{5}\right)}^{+}$	$y_{1},y_{2}$ $,y_{5}$	1	0	-	-	0	-
$g_{L^{k}\left(l_{6}\right)}^{+}$	$y_{1},y_{2}$ $,y_{4}$	1	1	-	0	-	-
$g_{L^{k}\left(l_{7}\right)}^{+}$	$y_{1},y_{2}$ $,y_{5}$	0	0	-	-	1	-
$\tau \left(x\right)$	/	4	6	3	3	3	2
$\mu \left(x\right)$	/	3	2	2	1	2	1
Voting value	/	3/4	2/6	2/3	1/3	2/3	1/2
Final result	/	1	0	1	0	1	0

**Table 2 sensors-23-08067-t002:** RAkEL evaluation indicators before and after improvement (*k* = 3, *m* = 34).

	Improved RAkEL	RAkEL
AP ↑	**0.304 ± 0.0045**	0.283 ± 0.003
Coverage ↓	**0.199 ± 0.005**	0.408 ± 0.025
Hamming Loss ↓	**0.037 ± 0.008**	0.041 ± 0.002
One Error ↓	**0.026 ± 0.002**	0.030 ± 0.003
Ranking Loss ↓	**0.015 ± 0.005**	0.031 ± 0.002

**Table 3 sensors-23-08067-t003:** Evaluation indicators of improved RAkEL and other methods.

	Improved RAkEL	CLR	BR	LP
AP ↑	0.304 ± 0.0045	0.27 ± 0.003	**0.305 ± 0.0003**	0.217 ± 0.012
Coverage ↓	**0.199 ± 0.005**	0.751 ± 0.01	0.715 ± 0.0382	0.441 ± 0.03
Hamming Loss ↓	**0.037 ± 0.008**	0.092 ± 0.002	0.086 ± 0.001	0.085 ± 0.016
One Error ↓	**0.026 ± 0.002**	0.061 ± 0.001	0.044 ± 0.0002	0.057 ± 0.009
Ranking Loss ↓	**0.015 ± 0.005**	0.053 ± 0.001	0.050 ± 0.003	0.029 ± 0.006
Comprehensive Ranking	**1.2**	3.8	2.4	2.4

**Table 4 sensors-23-08067-t004:** Actual fault data maintenance diagnosis results.

Monitoring Indicators	Criterion	Repair Method Diagnosis Probability			Diagnostic Accuracy
		C3	C4	C6	
BV (kV)	>50	34.8%	52.2%	13.0%	100.0%
Moisture content (mg/L)	>10	20.7%	55.2%	24.1%	93.1%
Acid value (calculated in KOH) (mg/g)	>0.01	38.5%	50.0%	11.5%	88.5%
Dielectric loss factor (90 °C)	>0.005	18.2%	50.0%	31.8%	100.0%
H_2_ (μL/L)	>10	21.1%	78.9%	0.0%	94.7%
C_2_H_2_ (μL/L)	>0.1	33.3%	40.7%	25.9%	88.9%
Total hydrocarbon (μL/L)	>10	20.0%	72.0%	8.0%	96.0%

## Data Availability

The data will be made available on request.

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
