# Peer review of "Improved RAkEL’s Fault Diagnosis Method for High-Speed Train Traction Transformer"

_sensors, 2023, doi:10.3390/s23198067_

Round 1

Reviewer 1 Report

- Section titles may be enhanced to show the paper flow better. For example, Section 2 could be titled Background and Related Work and combined with Section 3. Then, section 4 could explain the proposed enhancement over the vanilla RAkEL method. 

- Figure 6 & 7 are unclear; please choose colored patterns to differentiate the two results. 

- Discussion of the obtained results could be enhanced to clarify the main differences between vanilla RAkEL and the improved version. as example why the results compared only at k=3 and m=34? 

Overall, the paper provides a clear overview of the paper's scope, methodology, and findings. Refinements in terms of clarity, explaining technical terms, and highlighting the practical implications could further enhance its impact.

Reviewer 2 Report

Authors propose an improved RAkEL (Random k-Labelsets) high-speed train traction fault diagnosis method. Starting from a large amount of "sleeping" fault maintenance data accumulated by the railway department, taking a single fault record as an example, using specific monitoring values to build an instance support, using the fault phenomenon of the corresponding monitoring index as a label, and using the rescue algorithm to extract related faults, Improve the step of selecting the k labelset in RAkEL, mine and use the association rules between data and faults, and realize the fault detection of traction transmission. 

The results look encouraging and motivating. But some contents need be revised in order to meet the requirements of publish.

(1)The abstract should be improved. Your point is your own work that should be further highlighted.

(2)The parameters in expressions are given and explained.

(3)The manuscript's motivations should be further highlighted in the manuscript, e.g., what problems did the previous works exist? How to solve these problems? 

(4) In Section 1. Introduction, the main contributions of this paper should be further summarized and clearly demonstrated. 

(5) Figure 6 and Figure 7 are not clear, please revise them.

(6) The literature review is poor in this paper. I hope that the authors can add some new references in order to improve the reviews. For example, https://doi.org/10.1016/j.engappai.2023.106004ï¼›http://dx.doi.org/10.1109/TCSS.2022.3152091ï¼›http://dx.doi.org/10.1016/j.marstruc.2022.103181 and so on.

Extensive editing of English language required

Reviewer 3 Report

In this paper, the authors proposed an improved RAkEL (Random k-Labelsets) based high-speed train traction fault diagnosis method. The results show that the improved RAkEL diagnosis method has improved in multiple evaluation indicators. Overall, the paper is well written and organized with a proper length. The contributions as well as the quality are both good. In addition, there are some points that are not very clear and should be addressed in the revised version:

1.The description of the existing work should be shorter in Introduction section.  Furthermore, more descriptions of the proposed method are needed.

2. How to restrict the influence to the detected gas concentration caused by ambient temperature based on the proposed improved RAkEL  method. Please give some remarks here. 

3. Deep PCA is an important multivariate statistical analysis research issue which is widely applied on incipient fault diagnosis of High-Speed train traction system. The authors should supplement some results on this aspect, for example the following references had given significant design results:

[1] Deep PCA-Based Incipient Fault Diagnosis and Diagnosability Analysis of High-Speed
Railway Traction System via FNR Enhancement. Machines, 2023, 11(4): 475.

Minor editing of English language required
